# The Role of BDNF in Experimental and Clinical Traumatic Brain Injury

**DOI:** 10.3390/ijms22073582

**Published:** 2021-03-30

**Authors:** David Gustafsson, Andrea Klang, Sebastian Thams, Elham Rostami

**Affiliations:** 1Section of Neurosurgery, Department of Neuroscience, Uppsala University, 751 24 Uppsala, Sweden; dfsgustafsson@gmail.com; 2Department of Clinical Neuroscience, Rehabilitation Medicine, Uppsala University, 751 24 Uppsala, Sweden; andrea.klang@akademiska.se; 3Department of Clinical Neuroscience, Karolinska Institute, 171 77 Stockholm, Sweden; sebastian.thams@ki.se; 4Department of Neuroscience, Karolinska Institute, 171 77 Stockholm, Sweden

**Keywords:** BDNF, neurotrophins, neurotrophic factors, traumatic brain injury, neuroregeneration, neuroprotection, brain, neurons

## Abstract

Traumatic brain injury is one of the leading causes of mortality and morbidity in the world with no current pharmacological treatment. The role of BDNF in neural repair and regeneration is well established and has also been the focus of TBI research. Here, we review experimental animal models assessing BDNF expression following injury as well as clinical studies in humans including the role of BDNF polymorphism in TBI. There is a large heterogeneity in experimental setups and hence the results with different regional and temporal changes in BDNF expression. Several studies have also assessed different interventions to affect the BDNF expression following injury. Clinical studies highlight the importance of BDNF polymorphism in the outcome and indicate a protective role of BDNF polymorphism following injury. Considering the possibility of affecting the BDNF pathway with available substances, we discuss future studies using transgenic mice as well as iPSC in order to understand the underlying mechanism of BDNF polymorphism in TBI and develop a possible pharmacological treatment.

## 1. Introduction

Traumatic brain injury (TBI) is the leading cause of death worldwide, and in young adults under 35 years of age, the death rate is 3.5 times that of cancer and heart disease combined [1]. TBI is a dynamic condition with an initial primary injury triggering secondary events that can progress over hours, months, and even years. These events can be harmful such as ischemia, energy failure and inflammation, or beneficial such as the increased expression of neurotrophins and growth factors that promote neuronal survival and plasticity.

Neurotrophins, in particular brain derived neurotrophic factor (BDNF), have been identified to play a prominent role in the cellular events that occur in restorative processes following TBI, such as neuronal survival, axonal sprouting, and synaptogenesis [2]. Due to the important role of BDNF, its impact has been widely investigated in experimental TBI models and human studies [3]. It has also been shown that BDNF polymorphism influence the outcome after TBI, but the underlying mechanism has not been fully understood. As we will explore in this article, there is also a discrepancy in the results from experimental models studying BDNF expression regarding time post-injury, anatomical localization, and outcome. In this review, we aim to cover the findings in experimental TBI investigating BDNF and human studies exploring the effect of BDNF polymorphism in order to identify key knowledge gaps to direct future studies.

### 1.1. Brain Derived Neurotrophic Factor (BDNF)

BDNF is the most abundant neurotrophin in the brain [4,5], playing an important role in the survival, differentiation, synaptic plasticity, and axonal outgrowth of peripheral and central neurons throughout adulthood [2,4,6]. BDNF mediates its effect through the high affinity tyrosine kinase receptor (TrkB) [7]. An increased expression of TrkB mRNA has been detected in the site of injury following spinal cord injury and TBI [8,9]. BDNF also binds to a pan-neurotrophin receptor p75NTR, a member of the tumor necrosis factor receptor (TNFR) family [10]. The p75NTR receptor has been suggested to have a dual role both as a facilitator of Trk-mediated neuronal survival and as a regulator of neuronal cell death through the initiation of apoptosis. Accordingly, one of the drawbacks of treatments using BDNF analogs has been that they also affect p75NTR receptors mediating apoptosis. Thus, a TrkB selective drug or p75NTR inhibitor would be more beneficial.

#### BDNF Val66met Polymorphism

The BDNF single nucleotide polymorphism (SNP) in rs6265 produces a missense mutation, Val66Met (196G/A), which causes deficits in the activity-dependent secretion of BDNF [11]. The Val66Met mutation occurs in about 30% of the population in the US (met carriers/met+). When translated, the BDNF molecule contains three important domains: the signal peptide, the prodomain, and the functional BDNF molecule, all acting on different receptors. Mature BDNF binds to TrkB receptors and promotes long-term potentiation (LTP), cell survival, and dendrite formation; whereas proBDNF binds to p75NTR and activates long-term depression (LTD), apoptosis, and reduces dendritic complexity. This dual mechanism works antagonistically and regulates neuroplasticity. The Val66Met mutation results in accumulation of metBDNF in the soma while valBDNF accumulates in punctuate vesicles in dendrites. Neurons cannot secrete metBDNF in response to activity resulting in impaired plasticity in met carriers.

## 2. BDNF Polymorphism in Human Studies

The most studied polymorphism of BDNF in humans is rs6265 (Val66Met), in relation to normal cognitive function, effects on psychiatric, neurodegenerative and neuroinflammatory disease, and traumatic brain injury [11,12,13,14,15]. This has also been the case in TBI [16,17], although additional SNPs such as rs71244 [18,19], rs1519480, rs7124442 [20], and rs1153659 [21] have been identified to influence the outcome after traumatic brain injury. The SNP rs6265 of BDNF causes the substitution of valine to methionine and the prevalence of heterozygote or homozygote carriers of methionine is around 30% the population of the United States [22], but varies globally [23]. Most studies compare homozygote valine-carriers (Met-) to heterozygote and homozygote methionine-carriers (Met+). A minority of studies have three groups [17,24,25], separating heterozygote met+ from homozygote met+.

Among healthy individuals (mean age 36 y), carriers of the met-allele had worse episodic memory measured with n-back working memory test, as well as abnormal hippocampal activity assessed by functional MRI (fMRI), compared to Met- [11]. Similar results have been shown in another study with healthy individuals (mean age 30 y), also measuring episodic memory and activity of the hippocampus with fMRI during the test [26]. The met-carriers had 25% less activation of hippocampal regions and poorer performance at recognizing “new” and “old” information. Interestingly, there are indications that being a met-carrier may have cognitive benefits when aging. The Scottish Mental Survey 1947 included genetic analysis and long-term follow up and showed that that met-carriers had better preserved non-verbal reasoning skills at older age (mean age 79) [15], indicating that met-carriers may conserve cognitive function better than met-, during aging. The benefits have also been shown in the risk of developing Alzheimer’s disease among non-TBI individuals, where the risk was higher for val-homozygotes later in life but the reverse among younger subjects [14]. This is consistent with the theory of met+ having a positive impact on the preservation of cognitive function at older age. 

### 2.1. BDNF Polymorphism and Outcome after Mild and Moderate TBI

The effect of BDNFval66met polymorphism on the early cognitive outcome after mild TBI (mTBI) have been assessed in two different studies and shown a worse result for met-carriers. At a subacute follow up (one month) after mTBI, both the Met+ controls and the Met+ TBI-group had a slower reaction time in both simple reaction time (SRTRT—simple reaction time reaction time) and information processing speed (measured with Gordon Continuous Processing Test—CPT) compared to Met-patients and controls [17]. This correlates to another study where the participants were tested in five cognitive domains: attention, memory, language, visuospatial, and executive functions, immediately after recovery and at a six months follow up [27]. Overall, Met- had a better neurocognitive performance, without any significant difference between the two time points. However, met-carriers performed significantly worse at the memory test at six months, indicating a deterioration of some domain of cognitive function. 

Nevertheless, in the follow up of military veterans after mTBI in the chronic phase (mean 4.5 years), evaluating memory and executive function, the Met+ group with TBI had better results than their corresponding control group, whereas the Met- had the opposite result [28]. This could indicate that presence of the Met+ SNP result in a lower baseline of cognitive function but offer a protective quality of cognition post TBI.

The risk for neurodegeneration after TBI has been discussed and evaluated in different studies [29]. Severe TBI has been associated with a higher risk for neurodegeneration but during the last decade this link has also been proposed for mild TBI [30,31]. To find early signs of dementia, the volume of Hippocampus was evaluated with MRI in individuals with history of mTBI compared to controls, in relation to several BDNF SNP’s. Rs1153659 minor allele was correlated to a higher risk of reduced hippocampal volume after mTBI compared to controls but there was no difference for rs6265 [21].

In a study investigating the effect of headings in soccer players during six months on remyelination assessed by diffusion tensor imaging (DTI), it was shown that the met-carriers had significantly less remyelination compared to val-homozygot players [32]. This may explain part of the underlying pathophysiology in the poor functional outcome following repetitive mTBI in soccer players. There might be a gender difference as indicated by a Larson-Depuis et al. where female athletes with a history of concussion had a better olfactory function if they were met-carriers compared to val-homozygotes [25]. Olfactory function was proposed as a potential indicator of structural and functional injury after concussion, as well as a structure of the brain known to be able to regenerate and susceptible to BDNF. 

Cognitive outcome after TBI is multifactorial. Age and gender affect the outcome [18,19,33] and the severity and type of brain injury will affect cognition. Additionally, psychiatric factors can have consequences for cognitive function. Post-traumatic stress disorder (PTSD) has become increasingly discussed as a possible consequence to TBI [34], which in return can have an impact on cognitive function. One study reports of an increased prevalence of PTSD among Met+ patients with mTBI [24]. Another psychiatric factor that can interact with cognition is depression, which in return can be a consequence of TBI [35]. Met+ students reported a higher incidence of depressive symptoms after self-reported mTBI compared to Met- students with a history of mTBI [36,37]. The difference was more pronounced among female students. 

The gender difference related to depressive and anxiety related symptoms was further addressed in a study that followed those symptoms in a group of patients one and six weeks post mTBI [36]. This study evaluates a different BDNF polymorphism and found that men with t-allele, a different polymorphism from val66met, had a higher risk for depression after six weeks, while this was not the case for women. Patients with a history of mild to moderate TBI and a diagnosed depression responded differently to Citalopram, depending on BDNF rs6265, where the best responders were val-homozygotes [38].

### 2.2. BDNF Polymorphism and Outcome in Severe TBI

Only one study could be identified that investigated the effects of BDNFval66met polymorphism on early recovery after severe TBI [39]. Bagnato et al. assessed the cognitive function of 53 patients in vegetative state following severe TBI using Rancho Levels of Cognitive Function Scale. The test was performed at 1-, 3-, 6-, and 12-months post injury and no difference could be detected between met-carriers compared to Met-.

There are a few studies on long term outcome following severe, penetrating brain injury in American Vietnam veterans were correlation of BDNF polymorphism and cognitive outcome has been studied [16,20,40]. The follow up of these studies are 40 years after the TB I. Two of these studied the impact of BDNF polymorphism on IQ and executive function in veterans with penetrating TBI in the prefrontal cortex (PFC). Both found preserved preserved cognitive function in met-carriers with TBI compared to non-met-carriers [16,40]. In the third study all penetrating injuries were included with no preference of PFC [20]. In this study no significant difference was found between the groups for SNP rs6265 but two other BDNF SNP’s had significant effect on IQ, rs1519480 and rs7124442. This may indicate that there is a protective effect of BDNF polymorphism following injury and there might be a synergistic effect with different anatomical regions. 

Three studies have used a gene risk score based on “risk” BDNF SNPs (rs6265 and rs 7124442) [18,19,33], adjusted for age, to assess global outcome and mortality after severe TBI. Two studies showed a lower mortality during the first year (8–365 d) after TBI among the younger population (<45 and <48 y) with the low gene risk score (ie met- and rs7124442 t-homozygote), whilst the older population with low gene risk score had a higher mortality [18,33]. The third study had similar results where lower age had a higher rate of survival at six months with low gene risk score, though these results were to some extent influenced by levels of CSF cortisol as well [19].

### 2.3. BDNF Levels in CSF and Plasma after TBI

Three studies could be identified where BDNF levels were measured in plasma and/or CSF. Simon et al. analyzed the levels of BDNF in plasma at admission to the ICU, at mean 6.4 h post arrival to the hospital following severe TBI in 120 males (Glasgow Coma Scale (GCS) 3-8). There was no correlation between BDNF plasma levels and short-term fatal outcome (ICU mortality vs. ICU discharge) or between isolated brain injury versus multitrauma [41]. In 12 children with severe TBI levels of BDNF in CSF and plasma at 2 and 24 h post trauma, there was no correlation with the Glasgow Outcome Scale (GOS) at discharge [42], though there was a sharp peak of BDNF after head injury in all subjects. Another study including 315 patients with severe TBI showed a relation between BDNF levels in CSF in relation to levels of cortisol in CSF during the first week after the injury, and 6-month mortality [19]. Similarly, high CSF levels of BDNF daily sampled the first week post injury, were associated with an increased mortality (8–365 d post injury), whilst the acute mortality (0–7 d) was associated with low serum levels of BDNF [33].

The studies of the effect of BDNF val66met polymorphism vary in outcome measure, timing, and type of brain injury. Generally, the studies are small and most study population are smaller than 200 individuals. As a consequence, prospective studies are scarce and the results are heterogenous. The met/met prevalence in Caucasian population is low and therefore most studies group met-heterozygote and met-homozygote together for analysis.

## 3. Experimental TBI

In this section, we summarize the current knowledge of BDNF expression after traumatic brain injury (TBI) in different animal models, as well as the correlation between novel TBI treatment options and BDNF expression, and their effects on behavioral outcomes. In the section evaluating BDNF expression after TBI compared to sham, we chose to exclude studies which did not clearly state the location or time of BDNF analysis as well as those which did not report the data from sham animals.

### 3.1. Experimental Animal Models

More than ten different trauma models have been used to study BDNF expression after TBI (Figure 1) and the most chosen method was controlled cortical impact (CCI), followed by fluid percussion injury (FPI) and weight drop models. No study specifically analyzed whether different trauma models differentially impact BDNF expression. The model chosen was determined by which type of trauma was sought to be emulated, e.g., blunt force or penetrating injury. We attempted to draw conclusions regarding the different trauma types impact on BDNF expression but there were conflicting results in the reviewed material. As an example, Colak et al. utilized a controlled cortical impact model whereas Wang et al. utilized a fluid percussion model and both found that trauma increased BDNF mRNA expression in the first day post injury (DPI) in cortical tissue ipsilateral to the injury [43,44]. On the other hand, Boone et al. also utilized a fluid percussion model and found decreased BDNF mRNA expression in the ipsilateral hippocampus after 20 h but not significantly affected expression at 4, 8, 16 and 24 h after injury [45]. In summary, to draw definitive conclusions there is need for a study that specifically analyze the different models’ impact on BDNF expression.

### 3.2. Anatomical Regions of BDNF Analysis

In the reviewed material, the analyzed anatomical regions varied greatly. In order to compare and examine trends in the published data the anatomical regions were grouped together (Figure 2). Studies analyzing different regions, for example both ipsilateral and contralateral hippocampus (HC), were counted separately for each region analyzed. BDNF expression was most analyzed in the ipsilateral hippocampus and ipsilateral cortex, in 36 and 31 studies, respectively. In 7 studies the ipsilateral- and contralateral hippocampi were combined for biochemical examination and was labeled “bilateral HC”.

### 3.3. Assessment of BDNF Expression 

Both the mRNA and protein expression of BDNF were assessed. The most common method for mRNA analysis were in situ hybridization and quantitative real-time polymerase chain reaction (qPCR). For BDNF protein quantification Western blot and enzyme linked immunosorbent assay (ELISA) were commonly used (Table A2). In most of the reviewed material BDNF protein expression was not explicitly specified into mature-, pro- or preBDNF. When not otherwise specified, we assumed that the study refers to total BDNF protein expression. Also, unless otherwise specified, the studies examined rat or mouse tissues.

#### 3.3.1. Hippocampus

BDNF mRNA expression in the ipsilateral hippocampus was examined at 16 different timepoints and BDNF protein expression in the ipsilateral hippocampus was examined at 23 timepoints (2 h- 56 d) (Figure 3a). Three of the studies reported increased BNDF mRNA expression in the first 1 to 6 h after injury [44,46,47], whereas one reported a decreased mRNA expression at 20 h after trauma in both TBI and sham groups [45]. At all other subsequent time points, BDNF mRNA expression in the ipsilateral hippocampus was reported to be decreased or not affected compared to the sham (Figure 3a) [9,47,48,49,50,51,52]. Interestingly, BDNF protein expression did not show any significantly increased expression in the first day after trauma and showed decreased or not altered expression in the following weeks (Figure 3b). One study reported a non-significantly affected expression in the hours after trauma but found increased protein expression at 26 h after injury [46]. BDNF protein expression in the ipsilateral hippocampus was repeatedly reported as non-significantly affected by trauma at 5 h after injury as well as 1, 7, 11, 15, 21, 26, and 37 DPI. However, several studies reported a decreased protein expression at 4, 7, 8, 10, 13, 14, and 28 DPI [53,54,55,56,57,58,59,60,61,62,63,64,65,66,67,68,69]. The expression data from each date was usually from a single study per date, as well as almost exclusively from CCI or FPI trauma. There was no clear difference between trauma groups from the available data.

Hippocampus contralateral to the injury was analyzed at six timepoints for BDNF mRNA expression and three for BDNF protein expression, ranging from 1 h to 56 days. Rostami et al. report that BDNF mRNA expression was increased at 1, 3, and 14 days after trauma in the contralateral hippocampus, but not at the chronic phase measured at 56 days post-injury [9]. In the first day after trauma, Yang et al. reported an increased BDNF mRNA expression at 1, 3, and 5 h post injury. Two studies found that BDNF protein expression was non-significantly affected by trauma at timepoints 7, 21 [58], and 26 DPI [68].

#### 3.3.2. Cortex

The ipsilateral cortex displays a similar pattern of expression as the ipsilateral hippocampus. During the first day post injury BDNF mRNA expression was exclusively reported as increased [47,70]. Kobori et al. reported an increased BDNF mRNA expression using rt-PCR at 2, 6, and 24 h after injury, but no significant change at day 3 and 14 [71]. As seen in Figure 3e–f, the initial increase in ipsilateral BDNF mRNA was not accompanied by a decisive increase in BDNF protein expression. Contrarily, four studies reported that BDNF protein expression was decreased in the first day, and otherwise decreased or not significantly altered by trauma at time points 4, 7, 8, 25, 28, 30, and 35 DPI [57,69,72,73,74,75,76,77,78,79,80,81,82,83,84,85,86,87]. The exceptions were Cekic et al. who found that CCI induced an increase in mature BDNF protein after 24 h and 7 days [88], as well as Nagatomo-Combs et al. who analyzed the number of BDNF-protein expressing cells in rhesus monkeys at longer time points after trauma and found an increased number of BDNF-expressing cells at 1, 6, and 12 months after injury [89].

Once again, the contralateral side is not as well examined. Corne et al. examined the cortical mRNA-expression in the contralateral parietal lobe at 21 days post-injury and reported a decrease of all BDNF exons in the TBI group compared to sham [90], while Yang et al. found no significant change in BDNF mRNA specifically in the contralateral neocortex at 1, 3, and 5 h after lateral cortical impact injury compared to sham-injured animals [47].

To summarize, several studies found that BDNF mRNA expression increases in the ipsilateral hippocampus and cortex in the first 1–2 days after injury and that BDNF mRNA increases in the contralateral hippocampus in days 1–14 after injury. This increase in BDNF mRNA is not accompanied by a clear increase in BDNF protein in neither cortex nor hippocampi, however the data remains insufficient to draw definitive conclusions.

### 3.4. Behavioral Tests

Several studies examining the correlation between BDNF-expression and TBI performed behavioral tests to evaluate functional outcomes, as seen in Figure 4. In the reviewed material, the most common methods of functional evaluation were Neurological Severity Score (NSS), which includes a composite of motor, sensory, reflex and balance tests, as well as Morris Water Maze (MWM), evaluating spatial memory. Novel object recognition, examining non-spatial object recognition, rotarod, evaluating balance and coordination, Open field test, assessing locomotor behavior and anxiety, Elevated Plus Maze, evaluating anxiety levels and finally, the Beam walk test, which examines balance, were used in more than 5 of the reviewed studies. The group termed “Other” includes tests that were only used in one study, such as sucrose consumption test, swing test, and tonic-clonic seizure score. A closer description regarding the behavioral tests will follow in the next section as well as in Table A1.

### 3.5. Treatment of TBI

A total of 73 studies used various interventions to examine their effect on BDNF-expression and functional outcome following TBI, as seen in Figure 5. The most commonly investigated was the effect of exercise and diet, followed by stem cell and gonadocorticoid treatment. To review all treatments available in traumatic brain injury is beyond the scope of this review, and we have focused on reviewing exercise, diet and stem cell treatment as well as interventions affecting the intracellular pathways following BDNF-receptor activation. 

#### 3.5.1. Exercise

Exercise as a therapeutic method in traumatic brain injury was examined in 14 studies. Exactly what constituted exercise differed between the studies, but the most commonly used method was the running wheel. Some of the studies differed between voluntary and involuntary exercise, and some articles did not properly define exactly what type of exercise was performed. Out of those 14, 11 studies found that either pre- or posttraumatic exercise increased cerebral BDNF protein expression in the ipsilateral and contralateral hippocampus as well as in the ipsilateral cortex, as compared to non-exercised animals [58,59,62,63,64,66,67,68,69,71,78,91,92,93]. Two studies found that BDNF expression did not change significantly in rats exposed to exercise after fluid percussion injury at 7 or 11 days after injury, respectively [63,78]. Finally, Wu et al. found that exercise combined with a diet enriched with docosahexaenoic acid, a plasma membrane phospholipid, and increased BDNF protein expression compared to animals without enriched diet and inactive animals. Most of the studies found that exercise also improved functional outcome after injury. One study utilized an infrared-sensing running wheel system and trained the mice for three days before surgery and three weeks after. They found that the exercise improved short-term fear-aggravated memory and spatial memory tested by passive avoidance test and the Y-Maze test [69]. The same study also examined the *HSP20* gene, a gene expressing the heat shock protein 20 which is a chaperone previously known to be expressed after tissue stress and to be protective in e.g., Alzheimer’s disease. They found that silencing of the *HSP20* gene canceled the exercise induced enhancement. Three studies that examined exercise found a correlation between improved BDNF protein expression (and one study found that exercise increased BDNF mRNA expression) and improved cognitive function in the Morris Water Maze test [64,67,68,71]. Griesbach et al. showed that exercised rats performed better in the Morris water maze test after injury compared to non-exercised rats, and that injured rats that received an inactivating TrkB-IgG antibody did not benefit from exercise, further suggesting that exercise exerts its ameliorating effect through the BDNF pathway [67].

Da Silva Fiorin et al. and Zhao et al. were the only ones that examined the effects of pre-traumatic exercise. da Silva Fiorin et al. found that while FPI alone had no significant effect on BDNF protein expression, previous physical exercise induced an increase in BDNF protein expression in the hippocampus at 1 and 14 DPI. Zhao et al. found that running wheel exercise for 4 weeks before trauma increased BDNF mRNA expression in the ipsilateral cortex of both sham and injured animals compared to non-exercised animals. They also found that pre-traumatic exercise improves motor function in the beam walking test and cognitive function in Morris water maze and novel object recognition tests.

Griesbach et al, utilized a voluntary wheel exercise and examined functional outcome after lateral fluid percussion and found no significant effect on gross motor impairment. Shen et al. examined different intensity training models and examined functional outcome after severe controlled cortical impact and found that while there was no significant difference between trauma groups in the Neurologic Deficit Scores, low intensity exercised animals performed better in the MWM compared to control and high intensity exercised animals. Griesbach et al. utilized a lateral fluid percussion injury model with voluntary wheel exercise and found no significant effect of exercise on BDNF expression nor the result of beam walk test [59,68,78]. Hicks et al. found that fluid percussion injury significantly decreased function in NSS and MWM, and that post-traumatic exercise increased BDNF protein expression but had no significant effect on NSS or the MWM [92].

#### 3.5.2. Diet

A total of 15 studies examined different dietary treatment effects on traumatic brain injury in animal models. This included treatment with Astaxanthin (a sea-food derivated hermal remedy with antioxidant effects) [77], blueberry [51], caloric restriction [94], celery oil extract [82], curcumin [53,61], ethanol [95], Immunocal (a cysteine rich protein supplement) [96], procyanide [65], resolvin [97], trelahose [98], Vitamin E [60] and n−3 fatty acid treatment [55,99], or n−3 fatty acid deficiency [54].

Chandrasekar et al. examined the effect of acute ethanol intoxication in conjunction with trauma and found that trauma increased BDNF mRNA in the hippocampi bilaterally at 1 and 3 h after trauma compared to sham, and that the TBI-induced upregulation of BDNF was markedly decreased by ethanol pretreatment [95].

Ren et al. examined Resolvin, a docosahexaenoic (DHA) essential n-3 fatty acid derivate. The study examined BDNF protein expression in the hippocampus at 7 DPI and also found that TBI induced BDNF protein expression, and that Resolvin D1 further increased BDNF expression and ameliorated the cognitive effects of TBI in fear conditioning and beam walking tests [97].

Agrawal et al. found that FPI reduced the BDNF protein expression in the frontal cortex at 7 DPI specifically in animals exposed to n−3 fatty acid deficiency, but that n−3 fatty acid pretreatment prevented this. They also showed that n-3 fatty acid treated groups spent more time in the open arms of the elevated plus maze, indicated decreased anxiety [99]. Ji et al. showed that treatment with Astaxanthin improved BDNF protein expression at 7 DPI in the ipsilateral cortex, as well as faster NSS recovery and improved performance in the rotarod test [77]. Krishna et al. found that blueberry supplementation increased BDNF protein expression in the ipsilateral hippocampus at 14 DPI, as well as improved performance in the Barnes maze, however in the elevated plus maze no significant change was seen in either trauma or treatment groups [51]. Wu et al. examined the ipsilateral hippocampus at 4 DPI and found that dietary curcumin improved BDNF protein expression after trauma as well as performance in the MWM [53]. Additionally, they later showed that dietary curcumin also improved BDNF protein expression at 8 DPI and improved outcome in the beam walk [61]. Ignowski et al. found that treatment with Immunocal increased BDNF protein expression in whole brain lysate at 3 DPI, and also improved outcomes in beam walk, rotarod, and barnes maze tests [96]. Procyanides were examined by Mao et al. who found that treatment increased BDNF protein expression at 14 DPI in the ipsilateral hippocampus and improved MWM performance [65]. Finally, Aiguo et al. found that Vitamin E treatment increased BDNF protein expression 1 week after trauma in the ipsilateral hippocampus and improved outcome as tested by the MWM [60]. In summary, several dietary treatments seem to impact BDNF expression and there is a correlation between increased BDNF expression and improved outcome.

#### 3.5.3. Stem Cell Treatment

In the reviewed material, 9 studies examined stem cell treatments and the effect on BDNF expression after TBI. Mahmood et al. examined intravenous treatment with marrow stromal cells labeled with bromodeoxyuridine (BrdU). They found increased BrdU-positive cells in the perilesional regions, indicating the migration of marrow stromal cells (MSCs). Furthermore, they found that MSC treatment significantly increased BDNF at 8 DPI but not 2 or 5 DPI compared to vehicle. Finally, they found that the MSC-treated group had improved scores in mNSS and rotarod compared to control groups [100]. Mahmood et al. also examined long term recovery (90 DPI) and different doses of intravenous bone marrow stromal stem cells (BMSCs) treatment. They found that higher doses of BMSCs (4 × 10^6^ and 8 × 10^6^) significantly increased BDNF-protein levels compared to low dose (2 × 10^6^) and vehicle. They also found that the high and intermediate dose (4 × 10^6^ and 8 × 10^6^) improved NSS compared to low and vehicle treated groups. Finally, they found a dose-dependent increase in perilesional GFAP expression [101]. Feng et al. examined intravenous administration BMSCs and found that BMSC-treated animals had significantly increased number of cells expressing sex determining region Y (SRY) co-labeled with either neural nuclear antigen (NeuN) or glial fibrillary acidic protein (GFAP) in the ipsilateral cortex of rats compared with vehicle-treated animals, indicating that BMSCs migrated to the injured region and differentiated to neurons and astrocytes. Furthermore, they found that TBI alone had no effect on BDNF protein expression at 14 DPI the ipsilateral cortex, but that BMSC treatment significantly increased BDNF protein expression compared to both sham and trauma groups [81]. Deng et al. examined the interaction between BMSC treatment and stromal cell-derived factor-1 (SDF-1), which is a chemokine involved in the migration and survival of stem cells. Specifically, they examined posttraumatic micro injection of BMSCs cultured in solutions with and without SDF-1. They found that the number of BDNF-positive cells increased in the BMSC-treated group and further increased in the group treated with BMSC cultured with SDF-1. Furthermore, they found that the BMSC+SDF-1 group had better outcome in NSS and MWM tests compared to both BMSC without SDF-1 and vehicle groups [102]. Kim et al. found that BDNF protein increased in the ipsilateral hemisphere at 2 DPI, but found no significant expression change at 8, 15, or 29 DPI in TBI groups compared to sham. They also found that intravenous treatment with human mesenchymal stem cells (hMSCs) further increased BDNF expression at day 2 but had no significant effect at the other dates post injury. Although hMSC migration to the injured zone was confirmed by anti-human nuclei antibody staining at 2 DPI, the increase was transient and found to be decreased at 15 DPI. Additionally, there was only a small increase in NeuN or GFAP positive cells. Finally, they found that hMSCs improved outcome in the rotarod and mNSS tests compared to the vehicle treated TBI group [103]. Qi et al. examined umbilical cord mesenchymal stem cells (UC-MSCs) transplanted into the perilesional region and found that UC-MSCs increased BDNF protein expression at 2, 3, and 4 weeks, but not at 1 week, after injury compared to vehicle treated TBI. Furthermore, they found that UC-MSC treated group had an increased number of GFAP-positive cells as well as improved scores in NSS compared to vehicle [104]. Wang et al. found that intraventricular UC-MSC transplantation significantly increased the number of BDNF-positive and GFAP-positive cells compared to the control group. Additionally, they found that the UC-MSC-treated group had lower scores in NSS compared to the control [105]. Cheng et al. examined Wharton’s Jelly, which is an umbilical cord matrix including human umbilical cord-mesenchymal stem cells. They found no significant change in BDNF protein expression in the ipsilateral cortex at 14 DPI in sham compared to trauma groups, but that both BDNF protein and mRNA were significantly higher in the TBI group that received Wharton’s Jelly transplantation into the perilesional region compared to vehicle-treated rats [72]. Xiong et al. found that trauma decreased BDNF protein expression in the ipsilateral cortex at 7 DPI. They examined neural stem cells (NSC) from neonatal hippocampi incubated for neurosphere formation, as well as neurospheres derived from BDNF knockdown mice. They found that transplantation of NSCs into the perilesional region reversed the reduction of BDNF protein levels, and that BDNF knockdown neurospheres produced less BDNF and synaptophysin. In addition to this, they found that NSC-treated mice had decreased NSS compared to mice treated with the BDNF-KD NSCs as well as the vehicle treated TBI group. The NSC-treated group also had improved performance in the rotarod test compared to the vehicle-treated TBI-group. In conclusion, they found that NSCs transplantation increased BDNF expression and improved outcome in NSS and rotarod tests through BDNF-activation [79].

In summary, all the reviewed studies examining stem cell treatment found that several types of stem cells increased BDNF expression. Five out of nine studies did not include a sham group separate from vehicle or trauma. Regarding functional outcomes, the two studies examining umbilical cord stem cell transplantation together with the study examining Wharton’s Jelly transplantation found improved neurological severity scores in treatment groups compared to vehicle [72,104,105]. The study examining Wharton’s Jelly also found that treated rats spent more time in the correct quadrant and had shorter latency to find the platform in the MWM, as well as spent significantly more time exploring the novel object in the novel object recognition test. This indicates that Wharton’s Jelly transplantation improves spatial and object recognition memory after TBI in rats. The three studies examining marrow stromal treatment found improved NSS in treatment groups compared to vehicle groups, and two of them also found improved motor deficits in the rotarod test in marrow derived stem cell treated groups compared to control [81,100,101]. Marrow stem cells also improved outcome in both shorter escape latency times and number of platform crossings compared to control in the MWM, indicating improved spatial memory [102]. In the study examining human mesenchymal stem cell transplantation, they found improved outcome in NSS and rotarod tests in treated groups compared to control [103]. Finally, the study examining neural stem cells found that treated groups had improved NSS outcome as well as motor function in the rotarod test after trauma [79].

#### 3.5.4. BDNF Pathway Treatment

The number of studies examining direct intervention of the BDNF pathway in TBI are few, and this review included four studies. Sen et al. found that TBI decreased BDNF protein expression in the ipsilateral cortex 21 days following injury. Furthermore, they examined the Protein kinase-like endoplasmic reticulum kinase (PERK), a kinase in the endoplasmic reticulum activated by stress such as TBI, which mediates the downstream inhibition of translation. Previous studies have found that phosphorylation of PERK leads to an increased activation of CREB and thus the downregulation of BDNF. They found that a PERK antagonist GSK2656157 increased BDNF expression and improved cognitive performance in the Morris Water Maze test. This indicates that the inhibition of this pathway increased BDNF protein expression which could contribute to the improved performance in the MWM [83]. Alders et al. and Yin et al. examined BDNF fused with a collagen-binding domain, and BDNF expression in the ipsilateral cortex at 28 days after injury and found that BDNF was most increased in the mice treated with BDNF fused with the collagen-binding domain, followed by animals treated with only BDNF followed by TBI. They found no significant difference in BDNF expression between sham animals and injured mice [57,86].

BDNF have a short half-life and low blood brain barrier permeability, and one group used nanoparticles coated by surfactant, poloxamer 188 (PX), to increase BDNF concentration in target areas. They found that TBI increased BDNF protein expression in ipsilateral and contralateral hemispheres 4 h after injury. Furthermore, BDNF expression was increased ipsilaterally in animals treated with BDNF together with nanoparticles with and without PX compared to vehicle and BDNF without nanoparticle treatment groups. Contralaterally BDNF expression was only increased in the group treated with BDNF together with the combination of nanoparticles and PX. In the functional evaluations, they found a spontaneous improvement of NSS in day 1 to 6, with no difference between groups. However, at day 7 there was a significant improvement of NSS in the group treated with BDNF together with both nanoparticles and PX group compared to the other treatment groups. In the Passive avoidance test, the sham group and group treated with BDNF together with both nanoparticle and PX outperformed the other groups which performed no better than the non-treated animals [106].

#### 3.5.5. 7,8-DHF & EVT901

Recently, the synthetic flavonoid 7,8-dihydroxyflavone (7,8-DHF) was discovered following a screening for small molecules that could selectively activate BDNF receptor TrkB. This means that 7,8-DHF may cause similar effects, as BDNF in the brain, and be more therapeutically useful due to its better absorption and ability to cross the blood–brain barrier. The 7,8-DHF has shown an ability to promote the growth of these dendrites into synapses to help restore communication between neurons in animal models of cognitive decline.

In an experimental TBI model administration of 7,8-DHF prior to injury reduced cell death of neurons in the hippocampus. Reduced cell necrosis and apoptosis could also be seen upon administration of (7,8-DHF) after simulated TBI in adult mice [107]. Recently 7,8-DHF treatment was combined with exercise post-injury in rats and showed to promotes enhanced levels of cell metabolism, synaptic plasticity and increased brain circuit function [108].

Additionally, a selective antagonist of p75NTR, EVT901, was recently identified [109]. EVT901 inhibits p75NTR in vitro while increasing TrkA phosphorylation, blocks apoptosis and increases neurite outgrowth in neuroblastoma cells. Furthermore, treatment with EVT901 in TBI exposed rats reduced neuronal death in the hippocampus and thalamus, reduced long-term cognitive deficits, and reduced the occurrence of post-traumatic seizure activity.

These two newly discovered drugs showed no harmful effect in animal models and, offer a promising opportunity for complementary pharmacological treatment of TBI.

### 3.6. BDNF in Transgenic Animals

BDNF expression in transgenic animals after TBI is a novel field, and we have included a total of three studies. The results of these studies naturally vary from non-transgenic animals and because of this the results and methods of these studies have not been included in the previous graphs.

Giarratana et al. examined Val66Met-transgenic mice (Met+) and utilized a repeated mild TBI model using a lateral fluid percussion injury model. Giarratana et al. found that the total BDNF protein was decreased in Met+ injured animals in the ipsilateral cortex at 21 DPI, but that pro/mature-BDNF protein was increased in the ipsilateral hippocampus at 1 DPI compared to Met-. Furthermore, they found that Met+-animals had larger volume of inflammation compared to Val66Val at 21 DPI and that Met+animals had an increased activation of microglia in both hippocampal and cortical tissues at both 1 and 21 DPI. Met+ also have increased activation of Caspase-3+ cells (a marker for apoptosis) compared to Met- at 1 DPI, and have increased levels of FluorojadeC+ cells (a marker for neurodegeneration) compared to Met- at 1 and 21 DPI. Finally, they also found that Val66Met-injured animals had an increased number of phosphorylated tau+ cells (a marker for neurodegenerative pathology) compared to Met- at 1 and 21 DPI, and an increased number of GFAP+ cells in the ipsilateral cortex in Met+ compared to Met- at 21 DPI, but not 1 DPI, indicating increased astrocyte activation and risk of glial scarring [110].

Gao et al. utilized a cre/flox conditional knockout (KO) of BDNF which enables a site-specific knockout of BDNF in the granular neurons of the dentate gyrus of the hippocampus. In the flox/flox control animals, they found that TBI increased BDNF protein expression in the hippocampus. In the conditional KO animals, they found significantly decreased BDNF levels in the dentate gyrus in sham treated animals, and that TBI increased the levels of BDNF protein in the dentate gyrus of KO mice to a lesser extent that the increase of flox/flox control animals. Furthermore, they found significantly increased number of FJB+-cells in KO animals compared to injured flox/flox control animals, which indicates that the conditional knockout of BDNF leads to increased cell death in the dentate gyrus after trauma. Moreover, they showed that TBI injury significantly induces newborn neuron death 24 h following moderate TBI injury, and that the BDNF conditional KO further increases newborn neuron death in the dentate gyrus [111].

Cheng et al. studied thrombospondin-1 (TSP-1) KO animals after controlled cortical injury. TSP-1 is an extracellular matrix protein secreted by astrocytes in the brain and has been linked to several cerebral pathologies. Chang et al. found that in wild type (WT) animals, TSP-1 increased in the ipsilateral cortex at 6 h to 3 days, then returned to normal levels. Examining the relationship with BDNF expression, they found that TBI increased BDNF protein expression in both the contra- and ipsilateral cortex in WT at 21 days after trauma. However, in *TSP-1* KO BDNF increased only in the ipsilateral cortex and not in the contralateral cortex. This might hint at a *TSP-1* gene depletion-associated resistance of BDNF. Moreover, they found that measurement of synaptophysin (a marker for synapse quantification) showed no difference between KO and WT groups before TBI, but that TBI similarly significantly decreased synaptophysin in the contralateral cortex compared to sham and WT. There was no significant difference in synaptophysin expression in the ipsilateral cortex between groups. Furthermore, TBI increased extravasation in the ipsilateral hemisphere, which was significantly exasperated in *TSP-1* KO mice compared to WT. In the functional tests, *TSP-1* KO significantly worsened performance in NSS compared to WT post-TBI, indicating worse motor-sensor response. Wire grip and corner test was not significantly different in KO and WT groups and returned to normal at 10 DPI. In the MWM, *TSP-1* KO mice had increased latency to find the platform compared to WT, but no significant difference in entry times or target quadrant. *TSP-1* KO might worsen spatial memory recovery after TBI [112].

## 4. Discussion

The reviewed material is very heterogenous regarding the examined brain regions, temporal analysis of BDNF-expression after injury, type of trauma model, and functional tests, as well as whether a sham group was presented or reported. There is an urgent need for the standardization of experimental design in order to provide more reproducible results and solid conclusions. Nevertheless, there is an overall pattern of transiently increased BDNF-mRNA expression in the first day after trauma in the ipsilateral hippocampus followed by an ipsilateral decrease and a contralateral increase. Similarly, in the ipsilateral cortex, BDNF-mRNA increased in the first day after trauma, followed by a tendency of decreased expression.

Regarding the human studies, there is a similar need for standardization and larger cohorts. Generally, the studies are small, most have a study population <200 individuals and a control group has not always been used. Outcome measures differ among the studies, especially when evaluating cognitive function. Additionally, the time point for follow up varies between the studies and the access of prospective studies are scarce. The met/met prevalence in Caucasian population is low and therefore most studies group met-heterozygote and homozygote together for analysis which begs the question if the functional effect is the same and whether the met+ result in a lower baseline of cognitive function but offer a protective quality of cognition post TBI.

### 4.1. Human Induced Pluripotent Stem Cell-Models in TBI Research

As previously described, TBI is a heterogenous and complex condition involving multiple CNS cell types. Cellular interactions and subcellular processes follow temporal and spatial patterns that vary between affected individuals, and between different injuries. Accordingly, experimental TBI is usually studied using in vivo models, typically rodents, recapitulating many of the aforementioned features. However, certain aspects of TBI, such as the contribution of cell autonomous versus non-cell autonomous factors may also be studied in vitro, allowing for more mechanistic studies of isolated processes. In addition, a disadvantage with the current in vivo models is possible differences between human and animal cells regarding gene and protein expression or response to pharmacological interventions. These differences may underly some of the difficulties in translating results from basic research to clinical applications.

### 4.2. Potential Advantages with iPSC-Models

In line with the literature, we propose that in vitro models using neuronal cell types differentiated from induced pluripotent stem cells (iPSCs) from human subjects could be used as a complementary model, e.g., for studying cell to cell interactions, diffuse axonal injury (DAI), neuroinflammation, and the screening of neuroprotective drugs [113,114,115]. Advantages using iPSC-based models include that no experimental animals are required; the impact of genetic variations can be studied at the cellular level; ability to study human neurons, which are not accessible in live patients; and that pharmacodynamic and pharmacokinetic properties of potential drugs can be determined in the target human cell types. Moreover, subpopulations of neurons and glial cells of interest can be studied individually or in co-cultures. 

More advanced models include the use of brain organoids derived from iPSCs, which better resemble the three-dimensional environment in the brain and allow for more complex analyses [116,117], but entails challenges regarding data acquisition and analysis. Notably, these models may also recapitulate non-acute aspects of TBI, including aggregation of hyperphosphorylated tau and tar DNA-binding protein 43 (TDP43) [117], which has been linked to neurodegenerative processes such as chronic traumatic encephalopathy (CTE).

### 4.3. Studying the Impact of BDNF Val66met Polymorphism

We propose that studies of iPSC-derived neurons and glia from TBI patients with *BDNF* val66met polymorphisms could give clues about how this genetic variation influences, e.g., secretion and signaling of BDNF, synaptic plasticity and neuronal and glial response to injury. Moreover, such a model would be well-suited for pharmacodynamic and pharmacokinetic studies of the effects of the two potential neuroprotective compounds 7,8-DHF and EVT901.

However, since the val66met polymorphism has been linked to various neurodevelopmental and neurodegenerative disorders [118,119], we therefore propose that detection of TBI-associated phenotypes primarily requires the combination with an established in vitro model for TBI, such as scratch, blast, high intensity focused ultrasound, hypoxia or stretch injuries [113,117].

### 4.4. Considerations Regarding Translation to Humans

It should be considered that the monumental leap from patient to cells in a dish may result in subtle, artefactual or clinically irrelevant phenotypes [113]. Therefore, when designing such study, the hypothesis must be clearly defined and based on existing knowledge, rather than being a screening method for cellular phenotypes.

Moreover, it should be considered that single nucleotide polymorphisms (SNP) often result in subtle and multifactorial phenotypes, which may involve “multiple hits” in patients. Certain phenotypes associated with SNPs may therefore not manifest in vitro.

Since iPS cells derived from humans are genetically heterogenous, phenotypic differences between a patient line and a control line may be due to other factors than the ones intended to study. One approach to overcome this problem could be to use multiple control lines, but as proof-of-concept, one would modify the genetic variation in the patient line of interest using targeted gene correction in order to create an isogenic control line.

In summary, iPSC based TBI models could be useful in the studies of how genetic variations in the *BDNF* gene affects neuronal and glial function, and to evaluate new drug candidates, but should be used wisely in order to generate clinically relevant result.

### 4.5. Treatment of TBI and Future Research

Regarding the treatment of traumatic brain injury, several of the studies showed promising results and there is evidence of a positive correlation between increased BDNF expression and improved functional outcome, at least in animal studies. This is made especially clear in the cases where the positive effects of the treatment were cancelled by a BDNF antagonist, but unfortunately this was not often utilized in the reviewed material.

Traumatic brain injury is a global health issue with potentially devastating life-long consequences for the individual patient. Both injury and rehabilitation are very complex, and more research is necessary in order to understand the pathological mechanisms, and to provide novel treatment options of the primary injury. The treatment of the BDNF pathway could provide a novel treatment option in improving functional outcomes. Although treatment potential with the BDNF-molecule itself is limited because of low permeability of the blood brain barrier and a short half-life, an option could be TrkB-agonist treatment, such as 7, 8-dihydroxyflavone.

## Figures and Tables

**Figure 1 ijms-22-03582-f001:**
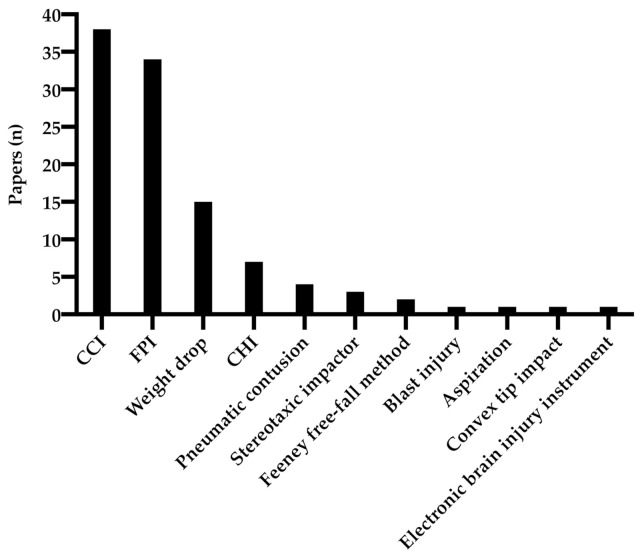
Trauma models in the reviewed studies.

**Figure 2 ijms-22-03582-f002:**
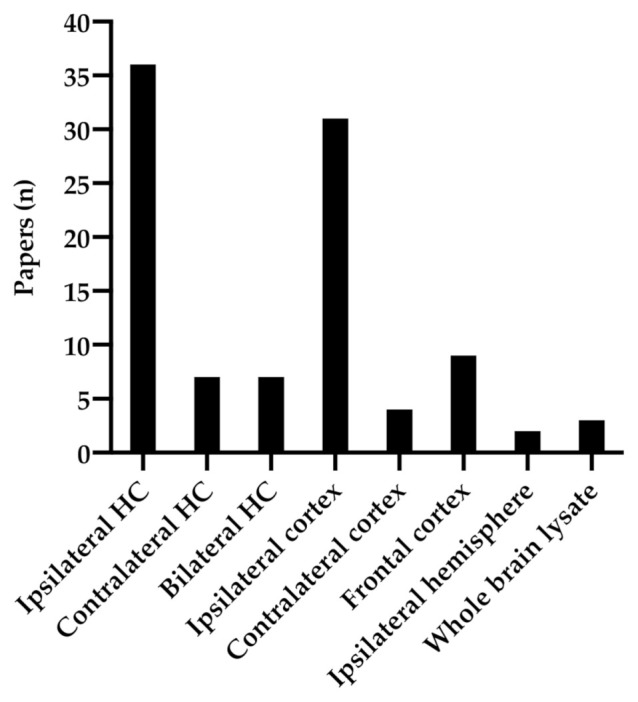
Anatomical regions analyzed in the reviewed material.

**Figure 3 ijms-22-03582-f003:**
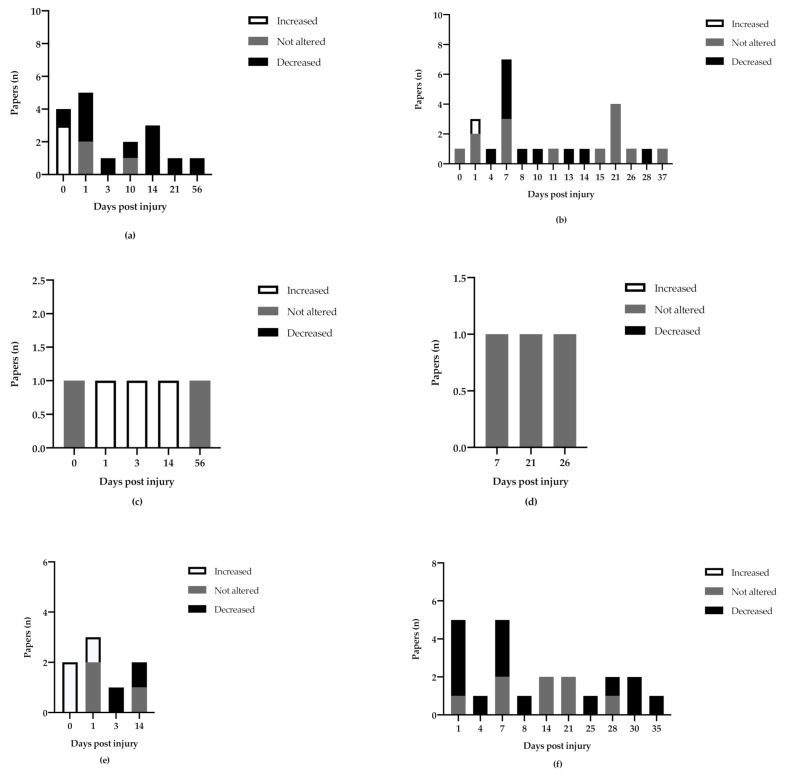
Overview of the number of timepoints in which brain derived neurotrophic factor (BDNF) expression was examined in the reviewed material, and whether TBI significantly increased, decreased, or was not significantly altered BDNF expression compared to sham-injured animals. (**a**) BDNF mRNA expression in the ipsilateral hippocampus. (**b**) BDNF protein expression in the ipsilateral hippocampus. (**c**) BDNF mRNA expression in the contralateral hippocampus. (**d**) BDNF protein expression in the contralateral hippocampus. (**e**) BDNF mRNA expression in the ipsilateral cortex (**f**) BDNF protein expression in the ipsilateral cortex.

**Figure 4 ijms-22-03582-f004:**
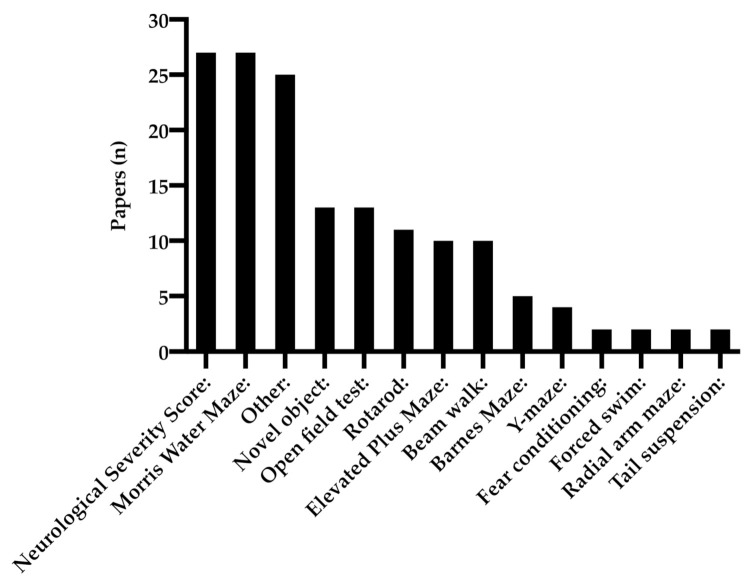
Behavioral tests analyzed in the reviewed material.

**Figure 5 ijms-22-03582-f005:**
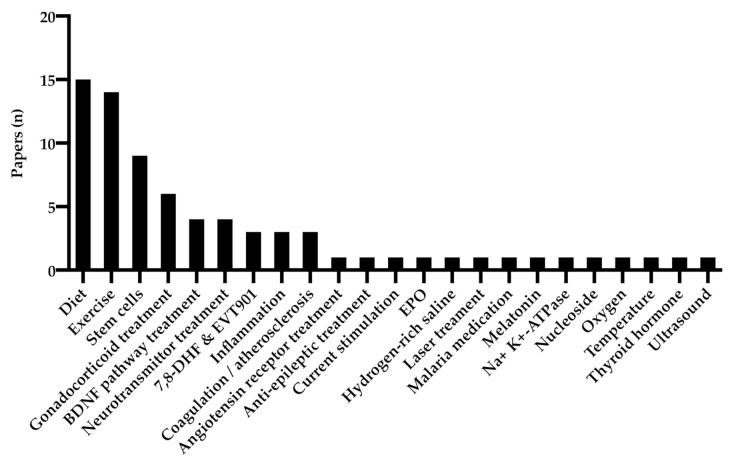
Treatment groups of the reviewed material.

## Data Availability

Not applicable.

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
