# Peer review of "The Role of BDNF in Experimental and Clinical Traumatic Brain Injury"

_ijms, 2021, doi:10.3390/ijms22073582_

Round 1

Reviewer 1 Report

In this manuscript, Gustafsson and colleagues review literature on BDNF in experimental and clinical traumatic brain injury (TBI). In particular, the authors discuss the association between BDNF polymorphism and brain function in humans either without TBI or after TBI. Moreover, the authors nicely summarize the changes of BDNF mRNA and protein levels in rodent models of TBI, as well as the potential strategy that can modulate BDNF expression levels/pathways. Each of these areas are subjects of intense study so this review is timely and relevant to the field. While the manuscript is well-written, I have a number of relatively minor concerns that should be addressed prior to publication.

Comments:

(1) Several abbreviations were not explained in the main text:

      >> What does GCS stand for in Line 177 “at mean 6.4 hours post arrival to the hospital following severe TBI in 120 males (GCS 3-8).”

      >> What does GOS stand for in Line 179-181 “In 12 children with severe TBI levels of BDNF in CSF and plasma at 2 and 24 hours post trauma, did not correlate with GOS at discharge”

     >> Line 142 “This study evaluates a different BDNF polymorphism and found that men with t-allele had a higher risk for depression after six weeks”. Does t-allele mean Val carrier or Met- carrier?

    >> What is DPI?

(2) Line 392 “They also showed that n-3 fatty acid treated groups spent more time in the open arms of the elevated plus maze, indicated decreased stress [98].” I feel like "decreased anxiety" would be more appropriate.

(3) Several typos need to be corrected. Please see the list below.

      >> Line 61 “Mature BDNF binds to TrkB receptors and promotes LTP”

      >> Line 153 “The test was performed at 1,3, 6- and 12-months post injury and no difference could be detected between met-carriers compared to and Met-.”

  >> Line 382 “found that trauma increased BDNF mRNA /in/ the hippocampi”

      >> Please check the references in line 537 and 541.

      >> Line 589 and 591, it should be synaptophysin.

Author Response

Thank you for the feedback and for the valuable pointers. Adjustments have been made to the text.

1) Several abbreviations have been explained, for example Glasgow Coma Scale (GCS), Glasgow Outcome Scale (GOS), Post-traumatic stress disorder (PTSD) and  days post injury (DPI).

2) In line 142 the t-allele represents a separate polymorphism from val66met. This has been clarified in the text.

3) In line 392 the text was changed to "decreased anxiety".

4) Several typos were corrected.

Reviewer 2 Report

The work is interesting. The authors try to clarify, in an area in which, despite a considerable scientific production, the results are not always comparable and sometimes conflicting with each other. In the final part of the paper, interesting ideas are proposed for future research.

I only have to report small inaccuracies that should be corrected:

  1. In the second part of the introduction from lines 36 to 41 some references should be introduced
  2. There is a large use of acronomics not previously stated. For example LTP, LTD, PTSD, GCS, GOS, FPI, MSC. Stating them would make it easier to read.
  3. Line 108 the adverb "however" is repeated consecutively in two periods. A synonym in the second would be better.
  4. On line 182 the authors write "n patients" why is the number not indicated? This is unclear to me.
  5. In section 3.5, the authors do not indicate any specific studies. I would insert a brief comment on the 2-3 studies considered most significant.

Author Response

Thank you for the feedback and for the valuable pointers. Adjustments have been made to the text.

1) References have been added to the introduction.

2) Several acronyms have been explained, for example Glasgow Coma Scale (GCS), Glasgow Outcome Scale (GOS), Post-traumatic stress disorder (PTSD) and days post injury (DPI).

3) On line 182 a synonym has been added instead of "however".

4) On line 182 the "n patients" have been replaced with the correct number "315 patients".

5) In section 3.5 we have attempted to clarify that a relevant in depth analysis will follow in the next section as well as a short summary in Table 1. As the graph summarised the results from all reviewed articles, we were unable to chose one particular article to reference.

6) Several typos were corrected.